# Somatotype Analysis of International Football Players with Cerebral Palsy: A Comparison with Non-Disabled Football Players

**DOI:** 10.3390/jfmk8040166

**Published:** 2023-12-11

**Authors:** Carmen Doménech, Enrique Roche, Raul Reina, José Manuel Sarabia

**Affiliations:** 1Sports Research Centre, Department of Sport Sciences, Miguel Hernández University of Elche, 03202 Elche, Spain; carmendomenechribes@live.com.mx (C.D.); jsarabia@umh.es (J.M.S.); 2Alicante Institute for Health and Biomedical Research (ISABIAL), 03010 Alicante, Spain; eroche@umh.es; 3Department of Applied Biology-Nutrition, Institute of Bioengineering, University Miguel Hernández, 03202 Elche, Spain; 4CIBER Pathophysiology of Obesity and Nutrition (CIBEROBN), Carlos III Health Institute, 28029 Madrid, Spain

**Keywords:** anthropometry, body composition, brain impairment, paralympic, para sport, adapted sport, soccer

## Abstract

Football for people with cerebral palsy is a para-sport involving ambulant athletes with impairments, such as hypertonia, ataxia, or athetosis. The objective of the present study was to describe the somatotype of a representative sample of international football players according to different functional profiles of cerebral palsy, including spastic diparesis, athetosis/ataxia, spastic hemiparesis, and minimum impairment criteria, and to compare it with non-disabled football players. A total of 144 international para-footballers and 39 non-disabled footballers participated in the study, and their somatotype was calculated using anthropometric measurements. A Kruskal–Wallis test was used to compare the groups to determine and assess the differences between the different functional profiles, and the analysis of anthropometric variables and body composition showed no differences. Regarding somatotype, a predominance of the mesomorphic component was observed in all subgroups, and differences in somatotype were also found between non-disabled footballers and para-footballers with spastic hemiparesis and minimum impairment criteria. This study suggests that there may be a degree of homogeneity in terms of somatotype among footballers with or without physical impairments, such as hypertonia, athetosis, or ataxia. Furthermore, it provides reference values of international-level para-football players for the different sport classes, which can help coaches and trainers monitor athletes’ physical conditions.

## 1. Introduction

Cerebral palsy (CP) is a non-progressive health condition that can manifest at various stages, including before, during, or after birth, leading to the loss or impairment of voluntary muscle control [1]. Its diverse symptoms can result in permanent disability due to brain damage affecting motor control [2]. Within the realm of sports for individuals with CP, seven-a-side football, governed by the International Federation of Cerebral Palsy Football (IFCPF), has gained prominence. This adapted sport integrates rules from the *Fédération Internationale de Football Association* (FIFA), involving a smaller playing field, the absence of offside rules, and the option for one-handed throw-ins with the ball below head height. Aligning with paralympic sports, CP football employs a classification system rooted in eligibility criteria. This system considers the presence and severity of deficiencies, such as spasticity, athetosis, or ataxia, and their influence on game skills, including passing, shooting, running, changing direction, or jumping.

One of the primary challenges in competitive sports for individuals with CP is the classification system, which categorizes para-athletes based on their impairments to ensure equitable competition. The aim is to highlight athletes’ anthropometric, physiological, and psychological attributes and how they utilize these, thus preventing disadvantageous competitive scenarios due to classification [3]. Players with eligible impairments that significantly affect their sporting performance meet the eligibility criteria for this para-sport. If an impairment is unclear or its impact on performance is not evident, the player’s ineligibility is categorized as “not eligible” (NE) [4].

The Classification Rules and Regulations of CP football [4] indicate that the eligible impairments primarily encompass neuromusculoskeletal and movement-related disorders [3], including spasticity, athetosis, and ataxia. Spasticity entails increased muscle tone, leading to constant muscle tension affecting speed and range of motion [5]. Ataxia presents as impaired control over voluntary movements, affecting balance and coordination [2]. Athetosis involves involuntary muscle contractions, typically resulting in slow, writhing movements in the hands, feet, arms, or legs [6]. In some cases, mixed muscle tone can occur, with some muscles too tense and others too relaxed, oscillating between hypotonia and hypertonia.

Additional considerations for players depend on the number of affected limbs. Monoparesis affects a single limb, while hemiparesis affects one half of the body and often presents with hypertonicity. In diplegia, frequently hypertonic, the lower extremities are affected, impacting static and dynamic balance and gait. Finally, tetraparesis encompasses all four limbs, with varying intensity in the upper and lower limbs, and may manifest as double hemiparesis when asymmetrical [7].

Moving away from the domain of cerebral palsy and sport, anthropometric studies play a pivotal role in evaluating athletes’ body composition and somatotype. These measurements offer valuable insights into their physical structure and serve various purposes, including tracking changes over time, comparing individuals, and analyzing the impact of specific pathologies [8].

The somatotype provides a quantifiable description of an athlete’s physique [9]. Coaches can leverage this information to tailor training programs, monitor rehabilitation progress, and achieve desired objectives. These studies encompass the estimation of body composition, morphology, dimensions, and proportionality through measurements taken at various anatomical sites and subsequently facilitate the calculation of the somatotype and somatochart [10]. The assessment of body composition is valuable not only for health-related applications, including nutritional status assessment in both healthy and diseased conditions, but also for studying body asymmetry resulting from certain pathologies [2].

The somatotype, which is viewed as a numerical representation of an individual’s morphological configuration, is expressed as endomorphy, mesomorphy, and ectomorphy components, listed in that order. Each component reflects distinct characteristics, with endomorphy indicating relative adiposity, mesomorphy representing relative muscular-skeletal robustness, and ectomorphy conveying relative linearity or thinness. On the other hand, the somatochart serves as a graphical representation of the somatotype [9].

The calculation of the somatotype is a versatile tool, allowing comparisons at different life stages and between individuals or groups. Additionally, individual somatotype analysis employs parameters, such as Somatotype Dispersion Distance (SDD) to compare with a reference somatotype. In addition, Somatotype Morphogenic Distance (SAD) dissects the components separately, providing insights into the magnitude of observed differences.

The somatotype and somatochart are useful in describing and comparing athletes at various competition or training levels; establishing connections with body composition data; characterizing physical changes during growth, development, and aging; aiding in gender comparisons; analyzing body image; and relating findings to specific pathologies and health indicators. Coaches can utilize this information to assess player progress over time, adapt training regimens according to intended goals, and monitor injury rehabilitation, among other applications [11].

Despite extensive research in the field of anthropometrics, studies on somatotypes in the context of CP football are limited. Existing research has focused on different sports disciplines, including swimming [12], sports for the visually impaired [13], athletics [14], and wheelchair sports [15]. However, these studies often involved mixed groups of para-athletes with diverse eligible impairments. In contrast, studies specifically addressing somatotype components in the context of CP football are scarce. From our knowledge, only three published studies have previously defined the somatotype components in this para-sport. Sarabia et al. [16] described the anthropometric profiles of CP football players according to their functional profile for the first time, although data from the somatochart were not included. Gomes de Macedo et al. [17] presented an anthropometric description considering only players with spastic hemiplegia, while Gorla et al. [18] described the somatotype of different sports classes. In both cases, the studies were performed with para-footballers at the national level.

To address these gaps in the specialized literature, the purpose of this study was to expand the existing knowledge about the somatotype profile and body composition of a representative sample of international para-footballers with CP. Specifically, this research aimed to consider different functional profiles based on the eligible impairment and its severity, offering valuable insights into the physical attributes and characteristics of these para-athletes and contributing to a more comprehensive understanding of somatotype within this specific para-sport. Moreover, the data provided by this work could aid coaches, athletes, and researchers in tailoring training regimens, monitoring progress, and enhancing performance within the context of CP football. By shedding light on the somatotype and body composition of these para-athletes, this research can be a valuable resource for both the scientific and sporting communities.

## 2. Materials and Methods

### 2.1. Participants

A total of 141 participants were involved in this study, encompassing individuals with CP and a control group (see descriptive data in Table 1). The selection of participants with CP was performed using a purposive sampling method, specifically targeting players actively involved in an international qualifying competition for the IFCPF World Cup. The CP group, with an average age of 25.9 ± 6.8 years, comprised 102 players from 12 different countries (45.5% of all the players from the 16 national teams that participated in that competition). They were categorized based on their CP profiles and severity [2], including spastic diparesis (*n* = 8), athetosis/ataxia (*n* = 14), spastic hemiparesis (*n* = 64), and those meeting the minimum impairment criteria (*n* = 16). The latter category referred to participants with a “minimal” alteration in muscle tone, postural control, and coordination compared to a “moderate” deficiency found in the previous profiles. In addition, a convenience control group (CG) was also established, comprising 39 players without disabilities actively participating in the third division of the Spanish football league. These players were recruited from football clubs in the province of the researchers’ center.

The selection of participants with CP was driven by the need to create a sample that accurately represented the international football landscape for this specific para-sport, ensuring a diverse representation of CP profiles and severity levels. On the other hand, the control group was strategically chosen to provide a relevant benchmark for comparisons, based on the total volume of training, to enhance the comparability and external validity of the study.

Before their involvement in the research, all participants provided written informed consent after receiving a comprehensive explanation, both written and oral, regarding the potential risks and benefits associated with their participation. This process adhered to the principles outlined in the Helsinki Declaration. The study received ethical approval from the Project Evaluation Office of the principal investigator university (Ref. DPS.RRV.01.14).

### 2.2. Anthtopometric Determinations

All variables were determined by a level 2 anthropometrist certified by the International Society for the Advancement of Kineanthropometry (ISAK) [19]. All measurements were made following the guidelines established by ISAK [20] with an individual technical error of 0.76–0.39% for skinfolds and 0.12% for the remainder of the parameters, representing acceptable errors for ISAK standards (<7.5% for skinfolds and <1.5% for the remainder of parameters). All measurements were taken twice, and the mean of the two measurements for each body side was calculated. If measurements had a difference greater than 1%, a third measurement was performed.

Each participant’s body mass was measured in kilograms using a digital scale (Tanita BC-601; Arlington Heights, IL, USA), breadths with a Holtain bone caliper (Holtain; Crosswell, UK), girths with a non-extensible metal tape (Lufkin; Missouri City, TX, USA), and skinfolds with a Holtain Tanner/Whitehouse skinfold caliper (Holtain; Crosswell, UK). Breadths from the humerus and femur were measured. Girths were measured on the relaxed arm and medial calf. Finally, four skinfolds were measured: triceps, subscapular, supraspinal, and medial calf.

### 2.3. Somatotype

The somatotype is a quantitative representation of an individual’s body shape and composition, encompassing three primary components: endomorphy, mesomorphy, and ectomorphy [9]. Each of these components provides distinct insights into an individual’s physical characteristics and plays a crucial role in understanding their body composition [17]. The three somatotype components were computed according to the Heath-Carter Anthropometric Somatotype Instruction Manual [21].

Endomorphy: Endomorphy quantifies the adiposity component of an individual’s body composition. It is calculated using the following equation:Endomorphy = −0.7182 + 0.1451 (X) − 0.00068 (X^2^) + 0.0000014 (X^3^)(1)

Here, X represents the sum of skinfolds from triceps, subscapular, and supraspinal sites in millimeters, multiplied by 170.18 and divided by the individual’s stretch stature in centimeters.

Mesomorphy: Mesomorphy characterizes the robustness and muscularity of an individual’s physique. The equation for calculating mesomorphy is as follows:Mesomorphy = 0.858 × humerus breadth + 0.601 × femur breadth + 0.188 × corrected girth from tensed arm + 0.161 × corrected girth from medial calf − stretch stature 0.131 + 4.5(2)

Ectomorphy: Ectomorphy relates to the linearity and leanness of an individual’s body. The calculation of ectomorphy is determined by considering the ponderal index (PI), which is defined as the ratio of stretch stature to the cube root of weight as follows: PI = stretch/^3^√weight.

Ectomorphy is computed based on the following formulas:

If PI > 40.75, then
Ectomorphy = (PI × 0.732) − 28.58(3)

If PI < 40.75 and >38.25, then
Ectomorphy = (PI × 0.463) − 17.63(4)

If PI < 38.25, a minimum value of 0.1 for ectomorphy is assigned.

The somatotype is expressed as a series of numerical figures representing the three components, listed in the same order, namely, endomorphy-mesomorphy-ectomorphy [21]. These figures are expressed with a decimal point and are rounded to the nearest unit of the mean value. Understanding the somatotype provides valuable insights into an individual’s body composition, with each component contributing to a comprehensive characterization of their physique.

### 2.4. Somatochart

The somatochart is a graphical representation used to visually depict and interpret an individual’s somatotype [9]. This two-dimensional diagram employs the X and Y axes to represent various somatopoints, which are essential in understanding the individual’s body composition and physique. The somatochart provides a unique perspective that complements the numerical representation of the somatotype, offering additional insights into an individual’s physical characteristics [17].

The key parameters defining the somatochart are as follows [9].

**X Axis**: The X axis represents the difference between ectomorphy and endomorphy. This dimension primarily focuses on the balance between leanness (ectomorphy) and adiposity (endomorphy) in an individual’s body composition. A higher value on the X axis indicates a more ectomorphic physique, while a lower value signifies a tendency towards endomorphy.
X = ectomorphy − endomorphy(5)

**Y Axis**: This dimension characterizes the muscularity and robustness (mesomorphy) relative to the individual’s adiposity (endomorphy) and leanness (ectomorphy). A higher Y value suggests a more mesomorphic physique, emphasizing the presence of muscle mass and strength.
Y = 2 × mesomorphy − (endomorphy + ectomorphy)(6)

### 2.5. Statistics

The following definitions were used to perform the comparative statistics of the somatotype [21].

Somatotype Attitudinal Distance (SAD): the distance in three dimensions between two somatopoints. It reflects the exact difference between two somatotypes. It is calculated using the values of the three components:SAD_A,B_ = √ [(ENDO_A_ − ENDO_B_)^2^ + (MESO_A_ − MESO_B_)^2^ + (ECTO_A_ − ECTO_B_)^2^](7)
considering ENDO = endomorphy, MESO = mesomorphy, and ECTO = ectomorphy. The subscript A corresponds to the somatotype studied, and B refers to the reference somatotype, in this case, the mean of the group. The higher the SAD values, the greater the difference between the somatotypes.

Somatotype dispersion distance (SDD): the distance between two somatopoints on the somatochart in two dimensions:SDD_A,B_ = √3 (X_A_ − X_B_)^2^ + (Y_A_ − Y_B_)^2^(8)
considering (X_A_, X_B_) as the coordinates of each participant obtained from the somatotype and (X_B_, Y_B_) as the mean value of the coordinates for the group.

Statistical analysis was performed using the JASP statistical package (JASP Team version 0.16; Amsterdam, NED). Statistical significance was set at the α level of 0.05. The data distribution was studied using Shapiro–Wilk tests and the Q–Q plot.

The results indicated that the data followed a normal distribution for the entire group as well as for the different profiles and severity of CP. Therefore, two multivariate ANOVAs (MANOVA) were conducted using groups (CP profiles and the control group) as independent variables and three somatotype components (i.e., endomorphy, mesomorphy and ectomorphy) or two somatotype distances (i.e., SAD and SDD) as dependent variables to avoid collinearity between variables. Significant associations were examined further using a univariate test, and, in the case of group interaction, Tukey’s honestly significant difference (HSD) test for multiple comparisons was used [22]. In addition, to calculate the effect size for pair comparisons between CP subgroups and CG, practical significance was assessed by calculating Cohen’s *d*, and the size of the effect is reported with the upper and lower 95% confidence intervals. Effect sizes greater than 0.8, between 0.8 and 0.5, between 0.5 and 0.2, and less than 0.2 were considered large, moderate, small, and trivial, respectively [23]. According to Carter [21], it should be considered that an effect size of 0.50 for the SAD mean difference is also considered relevant.

## 3. Results

MANOVA findings revealed significant associations between the groups and both somatotype components (Roy’s largest root = 0.557, F_(4, 136)_ = 18.94, *p* < 0.001) and somatotype distances (Roy’s largest root = 0.151, F_(4, 136)_ = 5.12, *p* < 0.001).

A predominance of the mesomorphic component was true across all subgroups (Table 1) when somatotype components of the players were analyzed. As shown in Figure 1, univariate testing revealed that there was a statistically significant difference in dependent variables between at least two groups for endomorphy (F_(4, 136)_ = 8.60, *p* < 0.001), mesomorphy (F_(4, 136)_ = 5.79, *p* = 0.001), SAD (F_(4, 136)_ = 5.10, *p* < 0.001) and SDD (F_(4, 136)_ = 5.00, *p* < 0.001). However, the interaction between groups and ectomorphy was non-significant (F_(4, 136)_ = 0.38, *p* = 0.824).

Specifically, Tukey’s HSD test for multiple comparisons between the CP subgroups and the control group is shown in Table 2. None of the comparisons between groups of CP footballers showed significant differences. These results demonstrated that the mean value of endomorphy was significantly different between CP subgroups, except athetosis players with the control group. For mesomorphy, only players with hemiparesis showed differences with CG. Furthermore, examinations of the spatial distance between somatotypes (i.e., SAD) and somatotype dispersion distance (i.e., SDD) uncovered notable distinctions between the CG and players with spastic hypertonia, both bilateral (i.e., diparesis) and unilateral (i.e., hemiparesis).

## 4. Discussion

This study aimed to enhance our understanding of the somatotype profile in international para-footballers with CP, comparing it with a comparable sample of able-body football players. The results revealed a consistent predominance of the mesomorphic component across all subgroups. Significant differences were observed in endomorphy, mesomorphy, SAD, and SDD among various CP groups and the control group. Players with hemiparesis showed the most different somatotype with respect to the control group. These findings highlight the nuanced somatotype variations within different CP profiles, emphasizing the need for targeted training approaches in CP football.

The somatotype is the quantitative description of the shape and body composition and is expressed in three components, namely, endomorphy, mesomorphy, and ectomorphy, representing the adiposity, robustness, and linearity of the body, respectively. Somatotype offers different applications, such as describing and comparing athletes, characterizing physical changes as a result of training, or relating aspects of these components with health indicators [24]. However, there are few studies describing the somatotype of international-level CP footballers in the different deficiency profiles due to the difficulty of obtaining data in large representative samples. Therefore, the main contribution of the present study is the description of the somatotype for each deficiency profile of elite CP footballers in a large sample of para-athletes at the international level.

The results of this study indicate that there are no significant differences between the anthropometric variables of the different functional profiles of CP football players. Regarding somatotype, a predominance of the mesomorphic component is observed in all functional classes, and this finding is similar to the results obtained by Gorla et al. [18] in a sample of Brazilian athletes of this para-sport. That study emphasized that this characteristic of the somatotype favors the performance of efforts with high muscular and speed demands. These variables are, in turn, associated with other sports demands, such as endurance and strength, determinants of performance in sprint actions, change of direction, jump, or fast and high-intensity efforts during the match [25] or small-sided game [26]. Given the observed mesomorphic predominance across all profiles of CP football players, coaches and trainers can strategically incorporate tailored strength and speed training programs. Emphasizing exercises that enhance muscular development and speed aligns with the somatotype characteristics identified in the study by Gorla et al. [18]. Specifically, targeted training modules can be designed to improve sprinting, change of direction and speed, and explosive actions, such as jumps. This application ensures a sport-specific training approach that aligns with the identified somatotype, potentially optimizing the players’ overall performance on the field.

However, when the components of the somatotype were independently analyzed, several differences were observed. In the subgroup with athetosis or ataxia, there was a balance between endomorphy and ectomorphy. However, in the remaining groups, a predominance of endomorphy versus ectomorphy was observed. This finding coincides with the finding obtained by Fernandes and Filho [27], where the studied players displayed a balanced mesomorph somatotype (2.46 − 4.97 − 2.69). Nevertheless, that study presented a small sample with no differences in functional profiles. The latter observation could be due to the fact that participants with spastic CP present increased muscle tone, causing persistent tension, while participants with ataxia do not display muscle hypertonia [7,28]. In our study, the predominance of endomorphy versus ectomorphy in most subgroups overlaps with the results obtained by Gorla et al. [18] for the whole sample, although this was not the case when differentiating between functional profiles.

The observed differences in somatotype components among CP football players subgroups suggest a nuanced approach to training and conditioning. For para-athletes with athetosis or ataxia, emphasizing a balanced regimen that considers both endomorphy and ectomorphy aspects could be beneficial. This may involve customized workout routines that address body composition nuances unique to this subgroup. Conversely, for players in groups where endomorphy predominates, targeted training to manage adiposity and enhance lean body mass may be particularly relevant. Tailoring training interventions based on these somatotype variations ensures a more precise and effective approach to meeting the specific needs of different functional profiles within CP football.

When comparing the different functional profiles with CG, we found significant differences in the endomorphy component for players with spastic hemiparesis and minimum impairment as well as in the mesomorphy component between the CG and players with spastic hemiparesis. This could be explained by different nutritional habits and degrees of training between groups [29]. The CG players were semi-professional footballers with a higher volume of complementary off-field training and, in some cases, controlled nutritional planning. Implementing targeted interventions can enhance the overall fitness and well-being of CP footballers. Coaches and sports nutritionists may consider developing personalized nutritional plans and off-field training programs for para-footballers. In addition, the players with CP came from different countries where, in some cases, this para-sport is not professionalized, or the recruitment of eligible players in the same geographical area for team training is complex, without prejudice to the possible barriers of access for the practice of complementary physical or sports activity derived from the disability itself [30]. Efforts to professionalize CP football in regions where it is not yet established could contribute to more standardized training practices and improved access to resources, ultimately benefitting the physical development of CP football players [31].

In this context, it should be noted that the result obtained by Gorla et al. [18] for mesomorphy of the subgroup with moderate spastic hemiparesis (3.87) is closer to that of our CG. Although it was a smaller sample of Brazilian para-athletes, it should be noted that this country traditionally ranks at the top of the world for this para-sport. Another possible explanation could be the heterogeneity of the players since, in our study, the players belonged to twelve different national teams, with a more homogeneous sample in terms of performance level compared to the players noted in Gorla et al. [18].

The analysis of body composition data indicated that the para-footballers participating in this study had a mean BMI of 22.62 ± 2.36 kg/m^2^, which is considered suitable for football players [32] and aligned, in turn, with that observed by Gorla et al. [18]. When analyzing this index according to the different functional profiles, we observed that players with minimum impairment criteria had a higher BMI. This result could be associated with the highest mean age of the participants given that age was previously recognized as an influential factor in weight gain [33]. The BMI obtained for most functional classes corresponds to that obtained by Gomes De Macedo et al. [17], but this study only had 10 players of the same class/functional profile (i.e., spastic hemiparesis).

Another aspect to consider is whether the somatotype or BMI obtained in elite CP football players, when used as a reference value for coaches and physical trainers of CP football players, could be susceptible to improvement. In this sense, the article by Sarabia et al. [16] demonstrated that despite the players presenting an adequate BMI, differences were found between the dominant and non-dominant sides in the subgroups with spastic hemiparesis and moderate ataxia/athetosis.

Some limitations of this study should be mentioned. First, although the number of CP footballers included in the study was a good representation of the international players of this para-sport, the sample size of some subgroups was different. This was due to the rules of the game when the study was conducted that established that each team had to have at least two players in FT5 (i.e., moderate spastic hemiparesis) or FT6 (i.e., moderate ataxia/athetosis) sport classes and a maximum of one player in the FT8 sport class (i.e., minimum eligible disability). This has resulted in an overrepresentation of those players with moderate spastic hemiparesis (i.e., FT7) [34]. Second, the equations used for the calculation of the somatotype were not specific for the different pathologies included in the study. Finally, it should be noted that, due to the fact that the data collection was performed before the start of an international championship with a cross-sectional descriptive design, no dietary records or hours of training of players with CP were available. Altogether, these nuances suggest that considering the modulating factors of the somatotype, such as the type and hours of training or the diet adapted to the pathology, would improve the profiles of reference. Finally, because this is a cross-sectional study with a convenience sample, it is not possible to infer causality, and data from females, a group of CP footballers competing at the international level since 2022, are missing.

## 5. Conclusions

In conclusion, the extensive examination of somatotype profiles and body composition in a substantial international sample of CP footballers revealed intriguing insights. Contrary to initial expectations, the somatotype profile and body composition did not exhibit significant variances among diverse functional profiles within CP football. However, compared to footballers without disabilities, these elite CP footballers showed some distinctions in their somatotype characteristics.

Moreover, the comprehensive insights gained from the somatotype profiles of international-level CP footballers offer tangible benefits for coaches and physical trainers. For instance, identifying the prevalence of the mesomorphic component highlights the potential for designing targeted strength and conditioning programs. Coaches can leverage this information to develop specialized training regimens that enhance specific muscle groups, fostering improved performance in activities, such as sprinting, changes of direction, and jumps. Additionally, the somatotype data can guide nutritionists in tailoring dietary plans to address the unique physical demands of CP football, optimizing energy levels and supporting muscle development. Overall, these findings offer practical applications for optimizing individualized training, nutrition, and performance strategies for elite CP football players.

## Figures and Tables

**Figure 1 jfmk-08-00166-f001:**
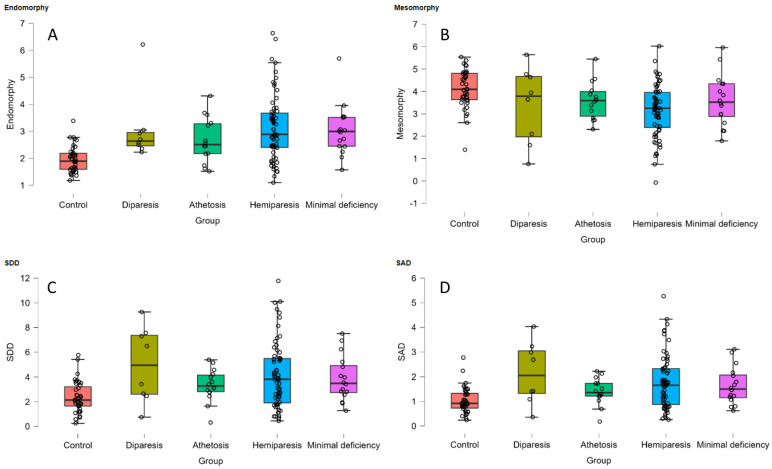
Interaction between groups and (**A**) endomorphy, (**B**) mesomorphy, (**C**) SDD (Somatotype Dispersion Distance), and (**D**) SAD (Somatotype Morphogenic Distance).

**Table 1 jfmk-08-00166-t001:** Descriptive sample characteristics.

Variable	Total Sample	Spastic Diparesis	Athetosis/ Ataxia	Spastic Hemiparesis	Minimal Eligible Deficiency	Control Group
*n*	141	8	14	64	16	39
Age (years)	24.83 ± 6.26	24.87 ± 7.45	26.07 ± 7.23	25.12 ± 5.96	29.12 ± 8.85	22.13 ± 3.29
Body mass (kg)	70.57 ± 8.44	65.56 ± 7.55	67.91 ± 7.22	68.28 ± 8.49	74.17 ± 8.22	74.85 ± 6.85
Stretch stature (cm)	176.16 ± 7.17	172.75 ± 5.17	173.71 ± 5.82	174.42 ± 6.88	177.56 ± 9.05	180.01 ± 6.04
BMI (kg m^−2^)	22.72 ± 2.16	22.02 ± 2.85	22.50 ± 2.03	22.44 ± 2.51	23.53 ± 2.06	23.07 ± 1.25
Endomorphy	2.7 ± 1.11	3.07 ± 1.30	2.68 ± 0.84	3.1 ± 1.23	3.03 ± 0.95	1.97 ± 0.48
Mesomorphy	3.5 ± 1.16	3.39 ± 1.72	3.60 ± 0.85	3.14 ± 1.17	3.60 ± 1.15	4.11 ± 0.86
Ectomorphy	2.7 ± 1.06	2.89 ± 1.49	2.66 ± 0.99	2.76 ± 1.22	2.43 ± 1.10	2.72 ± 0.64
SAD	1.53 ± 0.95	2.15 ±1.26	1.40 ± 0.55	1.76 ± 1.10	1.64 ± 0.76	1.03 ± 0.52
SDD	3.58 ± 2.30	4.99 ± 3.04	3.34 ± 1.35	4.12 ± 2.67	3.88 ± 1.83	2.36 ± 1.24
Somatotype Classification	Balanced mesomorph	Mesomorph- endomorph	Balanced mesomorph	Mesomorph- endomorph	Mesoendomorph	Mesoectomorph

BMI = body mass index, SAD = Somatotype Attitudinal Distance, SDD = Somatotype Dispersion Distance.

**Table 2 jfmk-08-00166-t002:** Post hoc comparisons between groups for somatotype components and distances.

			95% CI for Mean Difference					95% CI for Cohen’s *d*		
Dependent Variable	Comparison	Mean Difference	Lower	Upper	SE	*t*	Cohen’s *d*	Lower	Upper	*p* _tukey_
ENDO	Control vs. Diparesis	−1.10	−2.18	−0.02	0.39	−2.81	−1.09	−2.22	0.03	0.044 *
	Control vs. Athetosis	−0.71	−1.57	0.16	0.31	−2.25	−0.70	−1.60	0.20	0.169
	Control vs. Hemiparesis	−1.16	−1.72	−0.59	0.21	−5.65	−1.15	−1.76	−0.53	<0.001 ***
	Control vs. Minimal impairment	−1.06	−1.89	−0.23	0.30	−3.54	−1.05	−1.92	−0.19	0.005 **
MESO	Control vs. Diparesis	0.73	−0.455	1.91	0.43	1.70	0.66	−0.45	1.77	0.437
	Control vs. Athetosis	0.51	−0.43	1.46	0.34	1.50	0.47	−0.43	1.36	0.564
	Control vs. Hemiparesis	0.97	0.35	1.59	0.22	4.34	0.88	0.28	1.48	<0.001 ***
	Control vs. Minimal impairment	0.51	−0.40	1.41	0.33	1.55	0.46	−0.39	1.31	0.530
SAD	Control vs. Diparesis	−1.12	−2.09	−0.15	0.35	−3.18	−1.23	−2.36	−0.11	0.016 *
	Control vs. Athetosis	−0.37	−1.15	0.42	0.28	−1.31	−0.41	−1.30	0.48	0.684
	Control vs. Hemiparesis	−0.73	−1.24	−0.23	0.18	−4.00	−0.81	−1.41	−0.22	<0.001 ***
	Control vs. Minimal impairment	−0.61	−1.35	0.14	0.27	−2.25	−0.67	−1.52	0.19	0.167
SDD	Control vs. Diparesis	−2.63	−4.97	−0.29	0.85	−3.11	−1.21	−2.33	−0.08	0.019 *
	Control vs. Athetosis	−0.98	−2.86	0.90	0.68	−1.44	−0.45	−1.34	0.44	0.603
	Control vs. Hemiparesis	−1.76	−2.99	−0.54	0.44	−3.98	−0.81	−1.41	−0.21	0.001 **
	Control vs. Minimal impairment	−1.52	−3.31	0.27	0.65	−2.35	−0.70	−1.55	0.16	0.136

SAD = Somatotype Attitudinal Distance, SDD = Somatotype Dispersion Distance, SE = standard error, * *p* < 0.05, ** *p* < 0.01, *** *p* < 0.001.

## Data Availability

Data are available upon reasonable request to the corresponding author.

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
