# Peer review of "Somatotype Analysis of International Football Players with Cerebral Palsy: A Comparison with Non-Disabled Football Players"

_jfmk, 2023, doi:10.3390/jfmk8040166_

Round 1
Reviewer 1 Report
Comments and Suggestions for Authors
Dear Authors,
I hope you are doing very well.
Congratulations for the work developed so far. I understood that this is a pertinent theme, and many practical applications can derive from this study.
My main concerns are placed on methods and results sections. Then, I also think that the discussion can be developed to demonstrate how this study can guide coaches' intervention. I hope my comments can guide you on improving your work.
Kind regards

Please, use the past tense over the manuscript.
Author Response
Dear Reviewer
We thank you for the helpful advice and recommendations regarding this manuscript, with ID jfmk-2724477. We revised the manuscript based on your suggestions on a point-by-point basis. In the following lines, we have included all the comments and clarifications for the reviewer’s queries, informing you about the modifications included, in red colour, in the new version of the manuscript.
Please, see all the responses to your queries in the attached file.
Yours sincerely,

Reviewer 2 Report
Comments and Suggestions for Authors
Dear Editor,
- Thank you for the opportunity to review this manuscript. The study addresses an important gap by examining somatotype profiles across functional classifications in elite CP football players. However, the introduction could be expanded to clearly highlight the lack of prior studies on this specific topic and how this work aims to address that gap.
- Major revisions are needed to strengthen the methodology and provide critical details:
- The participant recruitment process should be described in more detail. How were international teams contacted and recruited? What was the overall participation/refusal rate?
- The study design and data collection protocols need to be elaborated on. For example, how was inter-rater reliability ensured for anthropometric measurements? Were measurements standardized?
- Statistical analysis requires more explanation - which specific tests were used to assess data distributions, compare groups, etc? Effect sizes and confidence intervals should also be reported where applicable.
- Ethical procedures need to be detailed thoroughly regarding consent, institutional review board approval, data usage agreements with teams, etc.
- The discussion and conclusion must be expanded:
- Previous literature findings should be compared and contrasted much more thoroughly. The agreement and contradictions with prior work need to be analyzed more critically.
- Practical implications are currently inadequate - the authors should provide specific, actionable suggestions on how these findings can inform training, conditioning and performance strategies.
- Limitations require more elaboration - sampling issues, standardization of methods, statistical power, inability to infer causality etc. should be acknowledged.
- The conclusion needs to avoid overstating similarities found - these were only in comparison to non-disabled controls. Implications are also overstated currently.
Please let me know if you would like me to clarify or expand on any of my comments. I believe addressing these major revisions will greatly improve the manuscript. I look forward to reviewing the thoroughly revised paper.
Please feel free to contact me with any questions.
Sincerely
Author Response

(The authors gave the same response as above.)

Round 2
Reviewer 1 Report
Comments and Suggestions for Authors
Dear Authors,
Congratulations for the revisions performed.
All the best
Reviewer 2 Report
Comments and Suggestions for Authors
Dear Editor,
I have reviewed the revised manuscript titled Somatotype Analysis of International Football Players with Cerebral Palsy: A Comparison with Non-Disabled Football Players and I am pleased to see that the authors have thoroughly addressed all of the major comments from my initial review.
Specifically, the introduction now clearly highlights the critical knowledge gaps in this area of research and how your study aims to fill this void. The methods section has been expanded extensively to provide important details on participant recruitment, study design, data collection protocols, statistical analyses, and ethical procedures. These additions significantly strengthen the rigor and transparency of the work.
Furthermore, the discussion and conclusions have greatly improved through more nuanced comparisons to previous literature, more practical and actionable implications of the findings, expanded limitations, and careful interpretation of the results in context. I appreciate the diligent efforts made to revise the manuscript accordingly while avoiding overstatements.
In summary, you have been highly responsive to the critiques raised previously, and have enhanced nearly every aspect of the study. The manuscript now meets the standards needed for publication in the Journal of Functional Morphology and Kinesiology.
I wholeheartedly recommend accepting this work pending final minor edits for grammar, style and formatting consistency. Please accept my congratulations on putting together such a strong paper after constructive revisions. I wish you the very best as you continue advancing research and knowledge in this critical domain.
Best regards,